# Vegetation Dynamics and Their Response Patterns to Drought in Shaanxi Province, China

**Jingxuan Su, Liangxin Fan \*** **, Zhanliang Yuan, Zhen Wang and Zhijun Wang**

School of Surveying and Land Information Engineering, Henan Polytechnic University, Jiaozuo 454003, China; sjx@home.hpu.edu.cn (J.S.); yuan6400@hpu.edu.cn (Z.Y.); wangzhen19@mails.ucas.ac.cn (Z.W.); 212104010036@home.hpu.edu.cn (Z.W.)
\* Correspondence: fanliangxin@126.com

**Abstract:** Effective water and vegetation management requires a better understanding of vegetation dynamics, and their response patterns to drought. Here, based on the normalized difference vegetation index (NDVI) and the standardized precipitation evapotranspiration index (SPEI), we investigate the vegetation response patterns to drought in Shaanxi Province (SAX), using Spearman's correlation analysis. The results show that the NDVI increased significantly ($p < 0.01$) from 2000 to 2019, with a trend of 0.054/10 yr. The maximum correlation coefficient between the NDVI and the SPEI ($R_{max}$) showed a significantly positive correlation ($p < 0.05$) in most areas (84.5%) of SAX. The $R_{max}$ of Northern Shaanxi (NS, median = 0.55) was higher than that of Central Shaanxi (CS, median = 0.53) and Southern Shaanxi (SS, median = 0.52). The corresponding timescale of $R_{max}$ was longer in CS (median = 7 months) than NS (median = 4 months) and SS (median = 3 months). The occurrence date of $R_{max}$ in NS (median = July) lagged that in CS and SS (median = May). The vegetation response patterns to drought varied with elevation; the $R_{max}$ decreased significantly ($p < 0.01$) with increasing elevation, while the corresponding timescale exhibited fluctuations. Additionally, Hurst exponent analysis indicated that 78.6% of the areas in SAX will exhibit a humidification trend in the future, and that vegetation growth in 74.7% of the areas in the region will be promoted.

**Keywords:** vegetation dynamics; drought; spatial patterns; regional heterogeneity; Shaanxi Province



## 1. Introduction

Drought occurs when the water availability remains below average levels for an extended period [1], leading to significant damage to terrestrial ecosystem vegetation. In the northern hemisphere mid-latitudes, drought causes a 48% decline in gross primary productivity (GPP) [2]. Similarly, the combination of drought and heatwaves in Europe in 2003 reduced the vegetation productivity by approximately 30% [3]. Additionally, the 2010 drought in south-western China caused a regional GPP reduction of 65 Tg C yr$^{-1}$ [4]. With climate change, the frequency and intensity of drought events are projected to increase, posing a severe threat to vegetation [5,6]. Therefore, studying vegetation response patterns to drought is vital to understanding vegetation vulnerability to climate change.

Numerous indicators have been developed for drought assessment [7]. Among them, widely used indicators include the Palmer drought severity index (PDSI) [8], the Standardized Precipitation Index (SPI) [9], and the Standardized Precipitation Evapotranspiration Index (SPEI) [10]. However, the PDSI has faced criticism due to its fixed timescale [11], given that drought has been widely accepted as a natural phenomenon with multiple timescales [1]. While the SPI considers multiple timescales, it overlooks the role of potential evapotranspiration (PET), a crucial component of the hydrological cycle [12]. On the other hand, the SPEI combines the advantages of both indicators, and provides a better characterization of drought events, particularly in arid and semi-arid regions [10,13]. In terms of vegetation assessment, the NDVI stands as the most commonly used indicator, utilizing the "red-edge" phenomenon to detect light and infer the presence of active plant

material and available water in the vegetation's root zone [14]. Therefore, the NDVI is an effective measure for characterizing vegetation response to drought.

　　By applying the SPEI and the NDVI, many studies have provided literature on vegetation response patterns to drought, and have demonstrated significant effects of drought on vegetation. Vicente-Serrano et al. [1] reported that the NDVI was significantly correlated with the SPEI in 72% of the global vegetated area. Zhang et al. [15] confirmed this conclusion, and found that the positive significant correlation between the NDVI and the SPEI was exhibited in most areas of China. Meanwhile, Xu et al. [16] showed that 43% of the vegetation in northern China was significant affected by drought. In addition to the response degree characterized by the correlation coefficient magnitude, researchers also focused on the timescale of the SPEI, which can describe the vegetation resistance to drought [1]. Qi et al. [17] reported that in the Qinling mountains of China, the forests responded to drought on a longer cumulative timescale, compared to the grassland and shrub. This conclusion has also been confirmed in the Yellow River Basin of China [18]. Based on the response degree and its corresponding SPEI timescale, recent studies examined the relationship between different vegetation types and drought. In general, woody plants can resist drought on a longer timescale, and with a lower response degree, compared to herbaceous plants [13]; this difference in the vegetation response to drought is primarily determined by the physiological properties of the vegetation type [1]. However, the vegetation response patterns to drought are not constant across regions, as they are also influenced by the topography, climate, and even human activities [19]. A study conducted in south-western China demonstrated that the response degree of forests was greater than that of grassland in the karst region [20]. Moreover, water requirements vary among vegetation types, and even within the same species, during different seasons [15]. Consequently, the vegetation response patterns to drought are not only related to the response degree and timescale, but also to the occurrence date that indicates the peak drought sensitivity in the vegetation [17]. Nevertheless, the response degree and corresponding timescale are considered, while the occurrence date are often neglected [13,21]. Therefore, it is necessary to study the vegetation response patterns to drought in specific regions, from multiple perspectives.

　　Owing to its dependence on the short-term precipitation events between June and August, drought is the most severe and frequent natural calamity in SAX [22,23]. From 1951 to 2012, the region experienced 110 episodes of moderate drought ($-1.5 < \text{SPEI} \leq -1$), and 18 cases of extreme drought ($\text{SPEI} \leq -2$) [24]. Moreover, the region has long suffered from severe soil erosion and vegetation degradation [25]. To mitigate this situation, the Chinese government identified SAX as a pilot province for the 'Grain to Green Project' (GTGP) in 1999, resulting in a sharp increase in vegetation coverage. However, research indicates that this policy may exacerbate the depletion of local soil moisture [26,27], thereby increasing drought sensitively in vegetation, while the vegetation response patterns to drought in SAX are still unknown.

　　Our primary objectives are as follows: (1) to assess the vegetation dynamics in SAX from 2000 to 2019; (2) to analyze vegetation response patterns to drought in SAX; and (3) to predict future trends in aridification/humidification, and the drought sensitivity of vegetation.

## 2. Materials and Methods

### 2.1. Study Area

　　SAX is situated in inland China (105°29'–111°15' E, 31°42'–39°35' N), covering a total area of approximately 205,000 km$^2$ (Figure 1a). The region is divided into three sub-regions: Northern Shaanxi (NS), Central Shaanxi (CS), and Southern Shaanxi (SS) (Figure 1b). Predominantly located on the Loess Plateau, NS covers an area of 92,521.4 km$^2$, accounting for 45% of SAX. The average altitude, annual temperature, and precipitation of NS are between 900 and 1900 m, 7 and 12 °C, and 400 and 600 mm, respectively. CS is one of China's significant grain-producing regions [28], and covers a total area of 55,623 km$^2$, with a vast central plain. The altitude, average annual temperature, and precipitation of CS are

190–3700 m, 14–16 °C, and 500–700 mm, respectively. SS, which encompasses a total area of 57,655.6 km², is mainly composed of mountains. The altitude, average annual temperature, and precipitation of SS are 300–2900 m, 15–17 °C, and 700–900 mm, respectively. By 2016, SAX had afforested a total area of 2.5 million ha, with NS, CS, and SS afforesting 1.123, 0.616, and 0.759 million ha, respectively [25]. From 2000 to 2019, the proportion of unchanged vegetation types in the region was 30.8% for cropland, 9.1% for forests, 11.8% for shrub, and 34.8% for grassland (Figure 1c). Based on local climatic characteristics, we define March to November as the growing season [29].

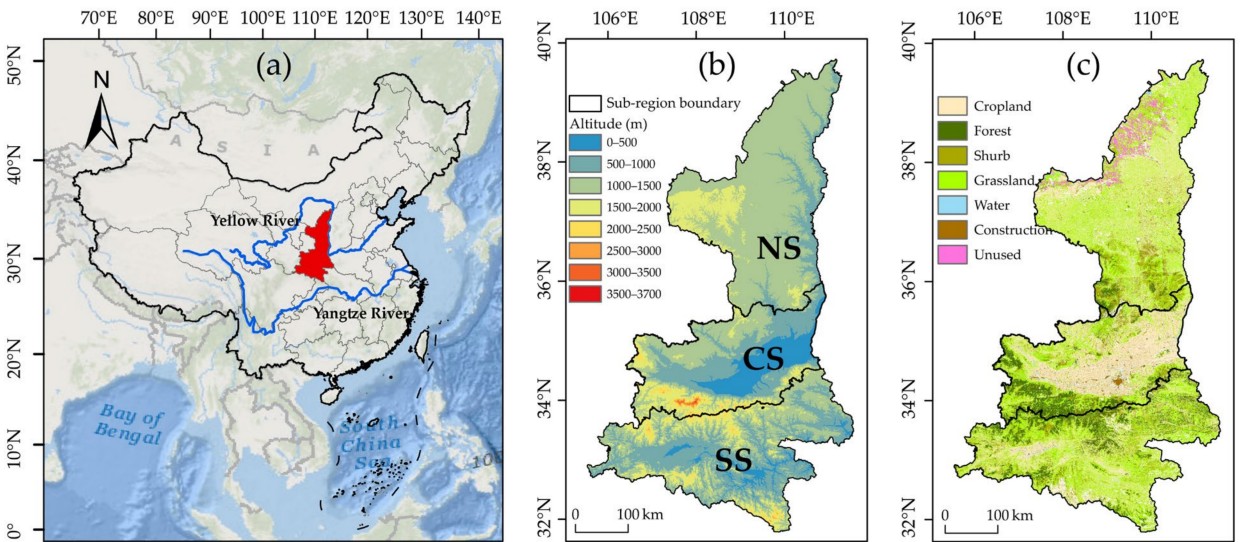

**Figure 1.** (**a**) The location, and (**b**) the topographical and (**c**) unchanged land-use types in SAX.

## 2.2. Data Collection and Preprocessing

### 2.2.1. Meteorological Data

The National Earth System Science Data Center provided monthly gridded precipitation and mean temperature data at a spatial resolution of 1 km. These data were obtained by downscaling the low-spatial-resolution climate data provided by the Climate Research Unit (CRU). A comparison between the downscaling results, and observations from 496 meteorological stations revealed deviations of 13.3 mm for the monthly precipitation, and 0.82–1.28 °C for the mean temperature [30].

### 2.2.2. NDVI Data

The Google Earth Engine (GEE) offers NDVI data (MOD13A2 V6.1) with a temporal and spatial resolution of 16 days and 1 km, respectively. These data have been widely used in vegetation monitoring, and have been proven to have a higher accuracy, compared to the GIMMS3g NDVI product [31]. To minimize the influence of clouds and atmospheric conditions [32], the acquired NDVI data have been processed using the maximum value composite (MVC) method on a monthly basis. The MVC method emphasizes the peak vegetation growth state. Its fundamental principle is to identify the maximum NDVI value among multiple NDVI observations for each pixel within a specific time range, which is then considered as the final NDVI value [33]. For instance, in March 2000, the MOD13A2 product provided NDVI data for two specific dates (5 March and 21 March). At the pixel level, we selected the maximum NDVI value from these two images, to represent the ultimate NDVI value for that particular month.

### 2.2.3. Vegetation Type and DEM Data

The Resource and Environmental Science and Data Centre provided us with vegetation type data at a resolution of 30 m. The data covered the period from 1980 to 2020; we downloaded the data for five years: 2000, 2005, 2010, 2015, and 2020. The elevation informa-

tion was derived from Global Multi-resolution Terrain Elevation Data 2010 (GMTED2010), with a spatial resolution of approximately 232 m. For detailed descriptions of each dataset, please refer to Table 1.

**Table 1.** Data description.

| Data | Data Source | Website |
|---|---|---|
| Temperature Precipitation | National Earth System Science Data Center | http://www.geodata.cn/ (accessed on 7 December 2022) |
| Vegetation type | Resource and Environment Science and Data Center | https://www.resdc.cn/ (accessed on 16 February 2023) |
| MOD13A2 NDVI GMTED2010 | Google Earth Engine | https://code.earthengine.google.com/ (accessed on 21 December 2022) |

### 2.3. Methods

In this study, we analyzed vegetation dynamics in SAX from 2000 to 2019, using NDVI data. Furthermore, we examined the relationship between the NDVI and the SPEI, through Spearman's correlation analysis. Finally, we employed the Hurst exponent, to assess the future drought sensitivity of vegetation. The technical flowchart for this study is shown in Figure 2.

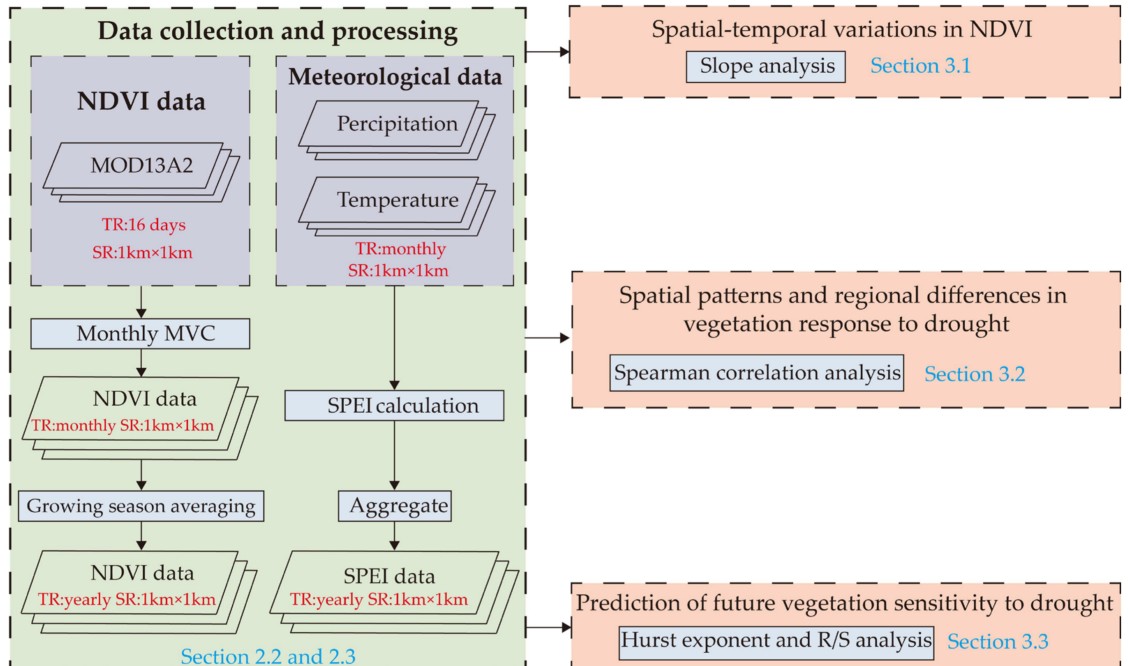

**Figure 2.** Flow chart for this study. Figure elements: TR, temporal resolution; SR, spatial resolution; MVC, maximum value composite; parallelogram, datasets; rectangle with blue background, study methods.

### 2.3.1. Slope Analysis

To detect the variation trends in the NDVI, we employed an ordinary least-squares-based linear regression analysis [34], with the following formula:

$$slope = \frac{n\sum_{i=1}^{n} i\mathrm{NDVI}_i - (\sum_{i=1}^{n} i)(\sum_{i=1}^{n} \mathrm{NDVI}_i)}{n\sum_{i=1}^{n} i^2 - (\sum_{i=1}^{n} i)^2} \tag{1}$$

where $n$ is the length of the NDVI series data, $\mathrm{NDVI}_i$ is the NDVI in the $i$th year, and *slope* > 0 or *slope* < 0 indicates the increasing or decreasing trend, respectively.

### 2.3.2. SPEI Calculation

The SPEI calculation process was divided into three steps. Firstly, we calculated the difference sequence between the monthly precipitation (*PRE*) and the potential evapotranspiration (*PET*):

$$D_i = PRE_i - PET_i \tag{2}$$

where $D_i$ represents the difference between the *PRE* and *PET* in *i*th month, $PRE_i$ represents the *PRE* in *i*th month, and $PET_i$ represents the *PET* in *i*th month. To calculate the potential evapotranspiration, we utilized Thornthwaite's approach [35].

Secondly, we fitted the $D_i$ sequence, using probability distribution. Ma et al. [36] employed the *Z*-test to compare five commonly used probability distribution functions (Weibull, Pearson-III, log-logistic, generalized Pareto distribution, and generalized extreme value distribution), and concluded that the three-parameter log-logistic distribution was more suitable for SPEI calculation in the Chinese region. Therefore, we selected a three-parameter log-logistic distribution to fit the $D_i$ sequence:

$$F(x) = [1 + (\frac{\alpha}{x - \gamma})^{\beta}]^{-1} \tag{3}$$

where $\alpha$, $\beta$, and $\gamma$ represent the scale, shape, and location parameters of the *D* values, respectively, and can be fitted using the linear moment method.

Finally, we determined the SPEI as the normalized value of *F(x)*, based on the classical approximation method [37]:

$$P = 1 - F(x) \tag{4}$$

$$W = \begin{cases} \sqrt{-2\ln(P)}, P \le 0.5 \\ \sqrt{-2\ln(1-P)}, P > 0.5 \end{cases} \tag{5}$$

$$\text{SPEI} = \begin{cases} W - \frac{C_0 + C_1 W + C_2 W^2}{1 + d_1 W + d_2 W^2 + d_3 W^3}, P \le 0.5 \\ \frac{C_0 + C_1 W + C_2 W^2}{1 + d_1 W + d_2 W^2 + d_3 W^3} - W, P > 0.5 \end{cases} \tag{6}$$

where $P$ represents the standardizing probability density function. $C_0$, $C_1$, $C_2$, $d_1$, $d_2$, and $d_3$ are constants. In this study, the SPEI with 1- to 12-month timescales were calculated using the "Geographic and Meteorological Analysis (GMA)" package in Python. SPEI = 0.5 and SPEI = $-0.5$ were taken as the threshold values for humidity and drought [36].

### 2.3.3. Correlation Analysis

Considering that the relationship between the NDVI and the SPEI may be nonlinear [16], Spearman's correlation coefficients between the SPEI and the NDVI were calculated for different months at different timescales, using the following equation:

$$R_{i,j} = Cor(\text{NDVI}, \text{SPEI}_{i,j}) \ 3 \le i \le 11, \ 1 \le j \le 12 \tag{7}$$

$$R_{\max} = \max\{R_{i,j}\} \tag{8}$$

where NDVI is the mean value during the growing season, *i* denotes the *i*th month from March to November, *j* denotes the timescale of the SPEI from 1 to 12 months, and $R_{i,j}$ denotes Spearman's correlation coefficients between the NDVI and the SPEI. Thus, 108 (9 × 12) correlation coefficients can be obtained for each pixel. Given that a larger correlation coefficient represents a greater impact of drought on vegetation, while a negative correlation implies an increasing NDVI value during the drought period (i.e., drought has no impact on vegetation growth) [38], we selected the maximum $R_{i,j}$ ($R_{max}$), which represents the highest drought sensitivity of vegetation, according to Equation (7). In this study, a two-tailed *t*-test was used to determine the significance level of $R_{max}$, with values of ±0.38, ±0.44,

and $\pm0.56$ indicating the 10%, 5%, and 1% significance levels, respectively. Spearman's correlation analysis was implemented using the "Pingouin" package in Python.

To further investigate the vertical patterns in the vegetation response to drought, we extracted $R_{max}$ and its corresponding SPEI timescale for different altitude intervals, in steps of 100 m, and fitted them using linear regression. In addition, we also extracted the $R_{max}$ and its corresponding SPEI timescale across vegetation types, using the "Extract by Mask" function in ArcGIS, to analyse the response patterns to drought under different vegetation types in SAX.

### 2.3.4. Hurst Exponent and R/S Analysis

To predict future trends in the SPEI, we utilized the Hurst exponent (H). To calculate H, we employed the widely used rescaled range (R/S) analysis [39]. The principles for the R/S calculation can be found in Sánchez Granero et al. [40]. The H ranges from 0 to 1, H > 0.5 indicates a consistent future trend compared to the past, H = 0.5 indicates the independence of future and past trends, and H < 0.5 indicates an opposite future trend compared to the past.

### 2.3.5. Drought Sensitivity Classification

The assessment of drought sensitivity for vegetation should be conducted from two perspectives: the response degree, and the corresponding SPEI timescale [17]. The response degree reflects the magnitude of the drought impact on vegetation, while the corresponding timescale indicates the vegetation resistance to drought [1]. Vegetation with a higher sensitivity to drought exhibits a greater response degree, and a shorter SPEI timescale, and vice versa [41]. Moreover, based on a two-tailed $t$-test, we have identified 0.38, 0.44, and 0.56 as the thresholds for positive correlation, at 10%, 5%, and 1% significance levels, respectively. Hence, we classified drought sensitivity for vegetation according to Table 2.

**Table 2.** Drought sensitivity classification for vegetation.

| Timescale | 1–6 Months | 7–12 Months |
|:---:|:---:|:---:|
| $R_{max} \leq 0.38$ | Not sensitive | |
| $0.38 < R_{max} \leq 0.44$ | Moderately sensitive | Mildly sensitive |
| $0.44 < R_{max} \leq 0.56$ | Severely sensitive | Moderately sensitive |
| $R_{max} > 0.56$ | Extremely sensitive | Severely sensitive |

## 3. Results

### 3.1. Spatial–Temporal Variations in the NDVI

Figure 3 shows the inter-annual variations in the NDVI in SAX. The multi-year mean NDVI is 0.53 during the growing season. In total, the NDVI values have exceeded the multi-year mean for 10 years, specifically in 2009, and from 2011 to 2019. The minimum NDVI value (0.45) and maximum NDVI value (0.58) occurred in 2000 and 2018, respectively, with a difference of 0.13. Overall, the NDVI in SAX showed a significant ($p < 0.01$) increase from 2000 to 2019, with a trend of 0.054/10 yr. In addition, we noted a sharp increase in the NDVI in SAX from 2000 to 2003, followed by a trend of steady, fluctuating increases.

The trend of NDVI variation in SAX spans from $-0.2/10$ yr to 0.23/10 yr, with 96.8% of the area exhibiting a positive trend. In NS, CS, and SS, the respective trends in the NDVI are 0.076/10 yr, 0.037/10 yr, and 0.042/10 yr. Notably, the central-eastern NS demonstrates the most pronounced increasing trend in the NDVI, while central CS has experienced a declining trend (Figure 4a). Moreover, a significant increase in the NDVI ($p < 0.05$) was observed in 98.9% of the SAX area from 2000 to 2019 (Figure 4b). In NS, CS, and SS, the percentage of areas with a significant increase in NDVI was 98.9%, 77.24%, and 87.9%, respectively. Overall, vegetation recovery was most prominent in NS, followed by SS and CS.

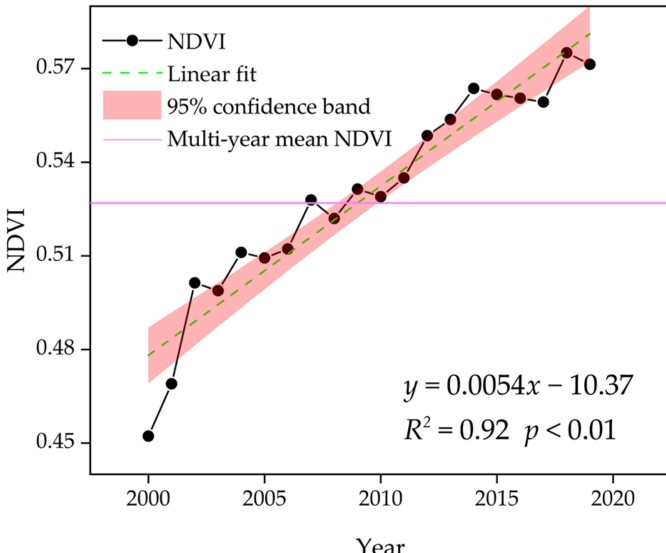

**Figure 3.** Inter-annual variations in the NDVI.

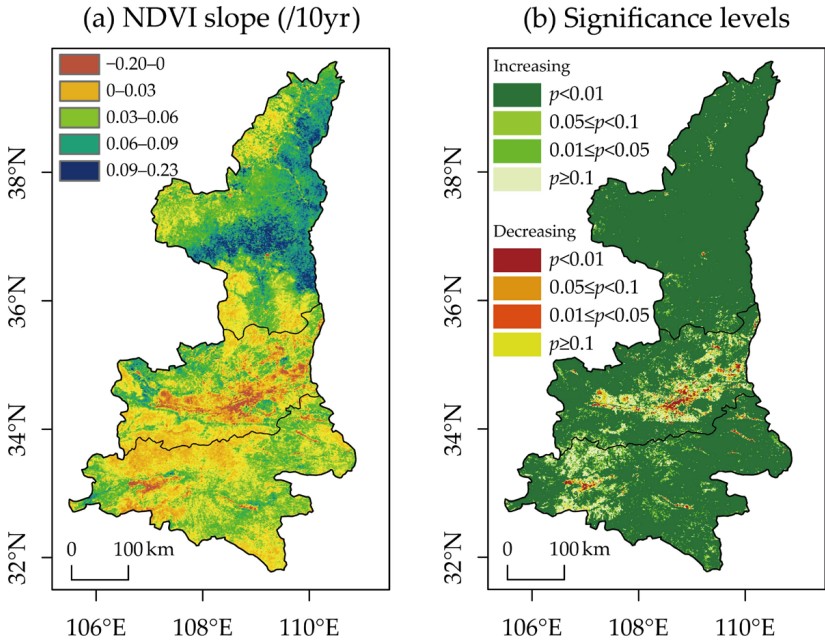

**Figure 4.** The spatial patterns in (**a**) the NDVI variation trends, and (**b**) the corresponding significance levels.

### 3.2. Drought Response Patterns and Their Regional Heterogeneity in Vegetation

3.2.1. Spatial Pattern of Response Degree

The vegetation in SAX exhibited a significant response to drought, with a median $R_{max}$ of 0.53 ($p < 0.05$) for the region during the growing season, from 2000 to 2019. The $R_{max}$ was statistically significant in 84.5% of the area in SAX, of which 38.5% and 45.9% were significant at the 1% and 5% levels, respectively. Additionally, the response degree in all the sub-regions was significant, with the median $R_{max}$ values of 0.55, 0.53, and 0.52 for NS, CS, and SS, respectively. Regarding the spatial distribution, the vegetation exhibiting a more pronounced response degree was mainly located in eastern NS, northern CS, and north-eastern and south-western SS (Figure 5a).

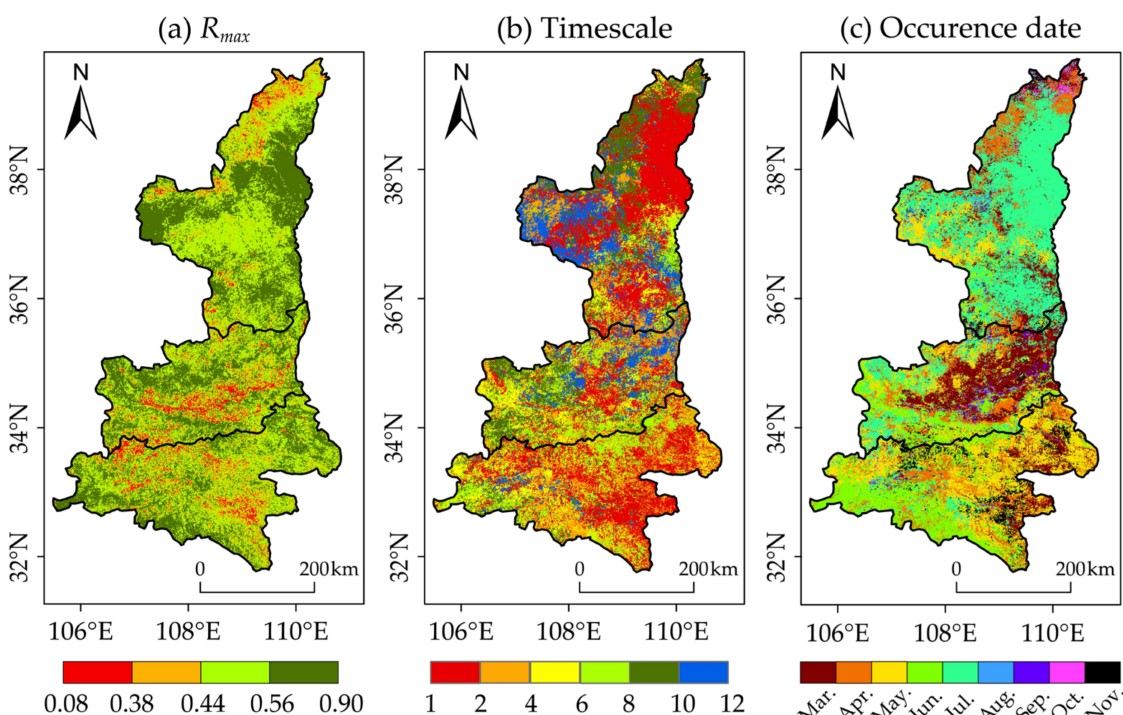

**Figure 5.** The spatial patterns of the (**a**) $R_{max}$, (**b**) corresponding SPEI timescale, and (**c**) occurrence date.

### 3.2.2. Spatial Pattern of Response Timescale

The response timescale provides an indication of vegetation resistance to drought, with short timescales indicating a low resistance, and vice versa. In SAX, the vegetation in 43.5% of the area was affected by drought on a 1- to 3-month timescale, with a predominant distribution in central-eastern NS and south-eastern CS. Meanwhile, the vegetation in 26.3% of the area exhibited the highest drought sensitivity on a 9- to 12-month timescale, mainly in the western areas of NS and CS. In CS, the vegetation exhibited the longest timescale of response to drought (median = 7 months), followed by in NS (median = 5 months), and in SS (median = 3 months) (Figure 5b).

### 3.2.3. Spatial Pattern of Response Occurrence Date

The occurrence date associated with $R_{max}$ reflects the period of vegetation most vulnerable to drought stress. In SAX, vegetation in 48.8% of the area was most sensitive to drought stress from March to May, primarily located in central-eastern CS and north-eastern SS, while vegetation in 43.5% of the area demonstrated the highest sensitivity to drought in June and July, mainly found in most parts of NS, western CS, and south-western SS. Overall, the vegetation in 92.3% of the area in SAX showed high sensitivity to drought from March to July. Additionally, the vegetation in NS was the last to respond to drought (median = July), while in CS and SS, the occurrence date of the vegetation response to drought was relatively consistent (median = May) (Figure 5c).

### 3.2.4. Vertical Patterns of Vegetation Response

We further analyzed the vertical patterns of vegetation response to drought in SAX (Figure 6). To mitigate the influence of human activities, we only selected unchanged vegetation types, within the altitude range above 500 m. As the altitude increased, the $R_{max}$ significantly decreased, indicating that the response degree diminished with higher altitude. Moreover, vegetation exhibited the highest response degree at around 1500–1700 m in altitude (Figure 6a). The response timescale varied with altitude, and could be divided into three intervals: 500–1700 m, 2700–3500 m, and 1700–2700 m. In the altitude range of

500–1700 m and 2700–3500 m, the response timescale increased significantly with altitude. However, in the altitude range of 1700–2700 m, the response timescale decreased significantly with increasing altitude (Figure 6b). Vegetation in areas above 3000 m showed a lower drought sensitivity, with a lower response degree, and a longer response timescale.

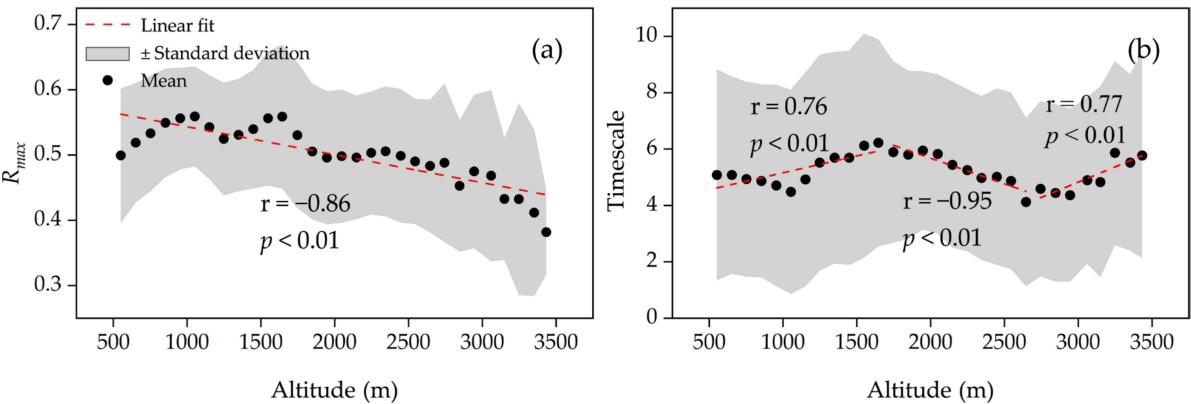

**Figure 6.** The vertical patterns of (**a**) the $R_{max}$, and (**b**) the corresponding timescale. Note: error band = ± 1 standard deviation.

### 3.2.5. Regional Heterogeneity of Vegetation Response

There was regional heterogeneity in the response patterns among vegetation types in SAX (Table 3). In NS, cropland exhibited the highest drought sensitivity, with a larger response degree and a shorter response timescale, followed by shrub and grassland. In CS and SS, grassland showed higher drought sensitivity, compared to shrub and cropland. Additionally, among all three sub-regions, forests exhibited a lower drought sensitivity, compared to the other three vegetation types. Overall, in SAX, grassland was most sensitive to drought, followed by cropland and shrub exhibiting similar levels of drought sensitivity, while forests show the lowest level of drought sensitivity.

**Table 3.** The median values of the $R_{max}$ and the corresponding SPEI timescales across vegetation types in different regions.

| | $R_{max}$ | | | | Timescale (Months) | | | |
|---|---|---|---|---|---|---|---|---|
| **Region** | **Cropland** | **Forests** | **Shrub** | **Grassland** | **Cropland** | **Forests** | **Shrub** | **Grassland** |
| NS | 0.56 | 0.53 | 0.55 | 0.54 | 3 | 5 | 5 | 5 |
| CS | 0.52 | 0.52 | 0.55 | 0.56 | 7 | 7 | 6 | 6 |
| SS | 0.53 | 0.52 | 0.54 | 0.54 | 4 | 4 | 4 | 3 |
| SAX | 0.54 | 0.52 | 0.54 | 0.54 | 5 | 6 | 5 | 4 |

### 3.3. Prediction Drought Sensitivity for Vegetation in the Future
#### 3.3.1. Spatial Patterns of Drought Sensitivity for Vegetation

According to Table 2, we classified the drought sensitivity of vegetation in SAX. We observed that vegetation in SAX is highly susceptible to drought stress during the growing season. Specifically, 24.1% of the area is classified as extremely sensitive to drought, while 42.1% is identified as severely sensitive. These areas are primarily located in north-eastern and southern NS, northern CS, and north-eastern and south-western SS (Figure 7a). Furthermore, when comparing the different sub-regions, we observed that SS exhibits a higher proportion of extremely and severely sensitive areas (69.5%), compared to NS (68.4%) and CS (58.8%).

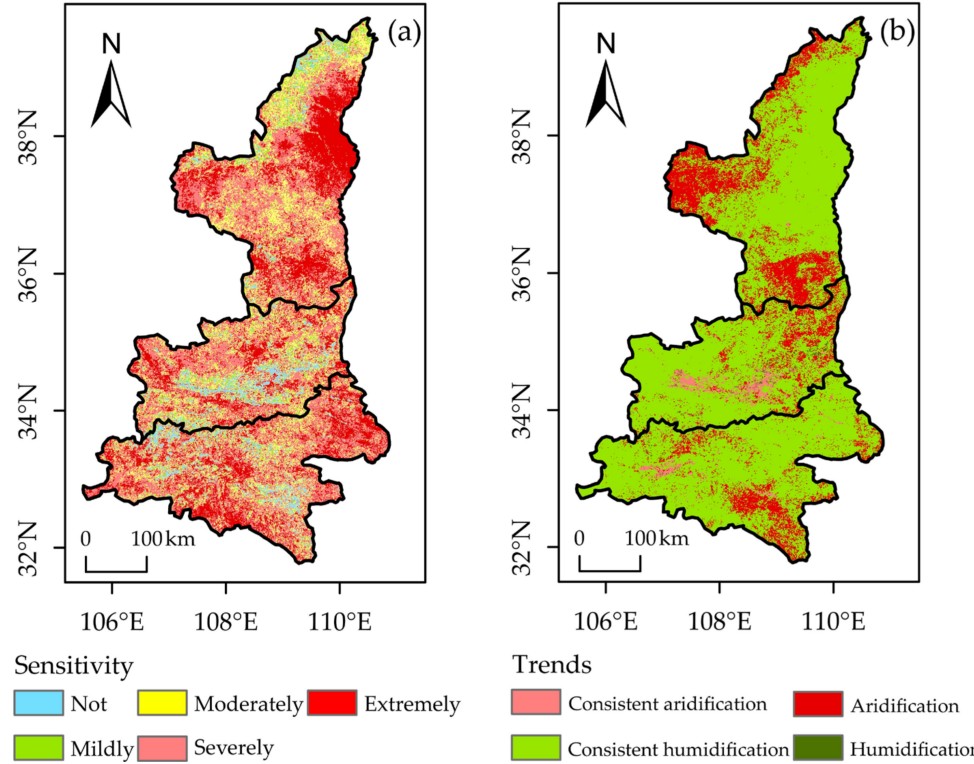

**Figure 7.** The spatial pattern of (**a**) the drought sensitivity in vegetation, and (**b**) the future trend of drought. Aridification refers to a condition where the current trend of drought is decreasing, but is expected to increase in the future, while humidification indicates that the current trend of drought is increasing, but will be alleviated in the future (H < 0.5). Consistent refers to a condition where the future trend of drought will align with the current stage (H > 0.5).

### 3.3.2. Future Trend of Humidification/Aridification

Based on the response timescale and response occurrence date corresponding to $R_{max}$, we predicted the future trend of humidification/aridification on the pixel scale (Figure 7b). In the future, humidification is predicted in 78.6% of SAX, and the humidification type will be dominated by consistent humidification, primarily in central-eastern NS, western CS, and north-eastern SS. Aridification is excepted to occur in 21.4% of the region area, primarily in western and southern NS, eastern CS, and southern SS. Moreover, we predict that 83.7% of the area in SS will show a humidification trend in the future, followed by CS (80%) and NS (73.4%).

### 3.3.3. Spatial Patterns of Drought Sensitivity for Vegetation in the Future

We overplied Figure 7a,b to obtain the classification of drought sensitivity for future vegetation in SAX (Figure 8). In the future, vegetation growth in 74.7% of the SAX area will be promoted by humidification, while growth in 19.3% of the area will be suppressed by aridification. Differentiating the sub-regions, future humidification is predicted to promote vegetation growth in 79% of the area in SS, followed by CS at 74%, and NS at 71.5%. Additionally, future vegetation management should focus on the western and southern parts of NS, the north-eastern part of CS, and the southern part of SS, where aridification is predicted, and vegetation is identified as being vulnerable to drought stress.

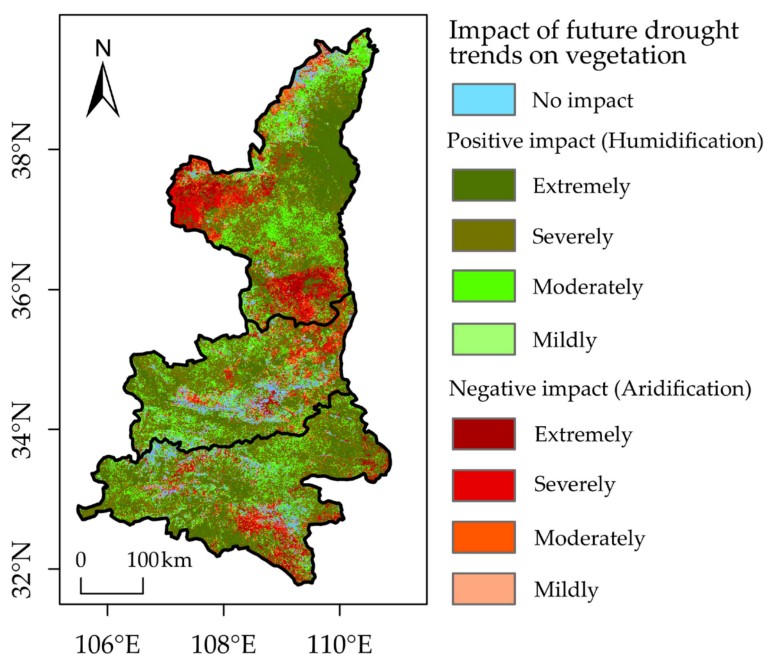

**Figure 8.** The spatial pattern of vegetation sensitivity to drought in the future.

## 4. Discussion

Consistent with numerous previous studies [13,15,16], our study revealed that the vegetation in most areas of the study region is vulnerable to drought during the growing season, emphasizing the importance of water availability as a major factor affecting vegetation growth in SAX. In NS and CS, vegetation exhibits a larger response degree to drought, compared to in SS. There are two possible factors contributing to this disparity. Firstly, NS and CS experience lower precipitation, compared to SS. Secondly, the "Grain to Green Project" (GTGP) has led to the significant depletion of soil moisture in NS and CS [42], which may further exacerbate the response degree. We observed that vegetation in NS and CS shows a longer response timescale, compared to in SS. There could be three possible reasons for this. (1) The entire NS and the majority of CS are situated on the Loess Plateau, the largest loess deposit in the world, and the soil has a high water-holding capacity [43,44]. As a result, the soil moisture changes more slowly in NS and CS, compared to in SS. This, in turn, contributes to the longer response timescale of vegetation to drought in NS and CS. (2) NS and CS are dominated by semi-arid regions, where vegetation tends to respond to drought on a longer timescale, due to its ability to tolerate water deficits [1]. (3) We observed that the vegetation in SS is more sensitive to drought around springtime. The rapid warming and inadequate rainfall in spring result in frequent occurrences of moderate and intense drought (SPEI $\leq -1$) in SS [45]. Chen et al. [46] reported that the SPEI $= -1$ is the warning threshold for most vegetation types. In other words, vegetation in SS is more vulnerable to drought with moderate and higher intensity around springtime. Ding et al. [47] confirm this conclusion, and found a higher correlation between the NDVI and the SPEI on a shorter timescale (1–3 months) in SS. Additionally, we also observed that the vegetation response lags behind in NS, compared to in CS and SS, possibly due to the higher latitude of NS. Huang et al. [48] illustrated that the optimum temperature for vegetation growth is $23 \pm 6$ °C, with a higher latitude meaning that this temperature is reached later, which correspondingly delays the critical vegetation growth period.

Previous studies have indicated a clear correlation and pattern between vegetation growth and altitude [49,50]. In SAX, we observed vertical variations in the vegetation response patterns to drought. Specifically, as the altitude increases, the response degree significantly decreases. The vegetation at altitudes above 3000 m exhibits the lowest response degree to drought. These high-altitude areas exhibit cold temperatures, low evapotranspiration, and high precipitation, creating cold and humid climate conditions.

Consequently, temperature and solar radiation conditions constrain the vegetation growth in such areas [51]. As the altitude decreases, the climate transitions from cold and humid to warm and dry, which alters the influence pattern of climate factors on the vegetation. In lower-altitude areas, water availability becomes the key factor controlling vegetation growth [50]. As a result, vegetation in these areas shows a higher sensitivity to drought. Furthermore, we observed that within the elevation range of 1700–2700 m, the rate of decrease in the vegetation response degree to drought slows down, accompanied by a shorter response timescale. This suggests an increased drought sensitivity in vegetation at this altitude range. One possible explanation is that within this altitude range, the effect of temperature on vegetation growth shifts from inhibition to promotion, amplifying the vegetation sensitivity to variations in the water balance, particularly a water deficit [52].

In SAX, there is regional heterogeneity in the response patterns to drought among vegetation types. Specifically, in NS, cropland exhibits the highest drought sensitivity. This can be attributed to the predominance of rainfed and mosaic cropland in the region [19]. Contrary to in CS and SS, the grassland in NS demonstrates a lower drought sensitivity compared to shrub. NS experiences the highest frequency of drought [45], making water availability crucial for vegetation growth. Compared to shrub, herbaceous plants require less precipitation to sustain growth [20]. Additionally, the ground coverage of herbaceous plants helps to reduce soil evaporation, and retain a small amount of precipitation during drought, contributing to their ability to maintain growth [53]. Li et al. [42] recommended that ecological restoration in NS should prioritize grassland, due to its ability to maintain the effectiveness and stability of soil moisture. In CS, a major food-producing region, we found that the drought sensitivity of cropland is comparatively low, compared to shrub and grassland. Human activities, such as irrigation and fertilization [54], have reduced the drought sensitivity of cropland. Shrub and grassland in CS can resist drought for a longer timescale, compared to NS and SS, while their response degree is more pronounced. Corresponding with the high evapotranspiration, a drought lasting for six months can deplete the soil moisture, and have a negative cumulative effect on vegetation growth [55], which may cause a significant decrease in the vegetation greenness [56]. Forests in SAX exhibit a lower drought sensitivity, which is consistent with previous studies [15,16]. Due to their deep root systems [57], forests can absorb water from deeper soil layers during drought periods, to maintain a normal physiological state. Therefore, compared to the other three vegetation types, forests exhibit the lowest drought sensitivity.

Please note the limitations of our study. We assume that the vegetation dynamics are mainly caused by variations in drought. In fact, the impact of pests, diseases, soil salinity and nutrient deficiencies, and human activities on vegetation cannot be ignored [34,58]. Furthermore, several studies have used solar-induced chlorophyll fluorescence (SIF), which characterizes vegetation photosynthesis, as an indicator to describe the physiological status of vegetation, and have suggested that SIF is more advantages for capturing drought stress [47]. Therefore, a future study should be based on multi-vegetation indicators, and focus on the combined effect of diverse factors, to further analyze the vegetation response patterns to drought.

## 5. Conclusions

In this study, we analyzed vegetation dynamics, and emphasized their response patterns to drought, in SAX, with the key conclusions drawn as follows. The NDVI in SAX experienced a significant increase from 2000 to 2019. Furthermore, a significant positive correlation was observed between the NDVI and the SPEI in most areas of the study region, suggesting that water availability plays an important role in vegetation growth. There is regional heterogeneity in the vegetation response patterns to drought among vegetation types in SAX. Therefore, it is important to implement differentiated management strategies for vegetation in different regions. In the future, the humidification trend is predicted in most areas of SAX, which will promote the growth of local vegetation.

However, attention should be given to the western and southern NS, and the south-eastern SS, where aridification is predicted, and vegetation is highly vulnerable to drought stress.

**Author Contributions:** Conceptualization, J.S.; methodology, J.S.; software, J.S. and Z.W. (Zhen Wang); validation, J.S., Z.W. (Zhen Wang) and Z.W. (Zhijun Wang); formal analysis, J.S.; investigation, J.S.; resources, L.F. and Z.Y.; data curation, J.S and Z.W. (Zhen Wang); writing—original draft preparation, J.S.; writing—review and editing, L.F.; visualization, J.S.; supervision, L.F. and Z.Y.; project administration, L.F. and Z.Y.; funding acquisition, L.F. All authors have read and agreed to the published version of the manuscript.

**Funding:** This research was funded by the Program for Innovative Research Team (in Philosophy and Social Science) at University of Henan Province (grant numbers 2022-CXTD-02 and 2021-CXTD-08), National Natural Science Foundation of China (grant number 42171299), and the Scientific and Technological Innovation Team of Universities in Henan Province (grant number 22IRTSTHN008).

**Data Availability Statement:** The data used in this study can be downloaded from the website given in Table 1.

**Acknowledgments:** We thank the National Earth System Science Data Center, the Resource and Environmental Science and Data Center, and Google Earth Engine for providing data support. We would also like to express our gratitude to the open-source Python packages Geographic and Meteorological Analysis (GMA) and Pingouin for providing algorithmic support.

**Conflicts of Interest:** The authors declare no conflict of interest.

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
