# Peer review of "Vegetation Dynamics and Their Response Patterns to Drought in Shaanxi Province, China"

_forests, doi:10.3390/f14081528_

Round 1

Reviewer 1 Report

Dear authors. I am glad to review the article "Vegetation dynamics and its response patterns to drop in Shaanxi Province, China". While reading the article, I had some questions.  In general, the article is devoted to a topical topic and will be very interesting to the world scientific community.

1. Provide a broader overview of the problem under study in section 1.

2. In Figure 1, specify the coordinate grid on the map of China, sign the seas, oceans, neighboring countries, the names of the designated rivers.

3. Use one citation style – [] (do not use - Granero et al. (2008)) 

4. The methodology is not entirely clear. Have you used GIS or GEE to perform calculations? Specify in detail the programs that you have used. If you used GEE, please provide the code for which the data was received in the additional materials.

Reviewer 2 Report

Dear Authors,

This study assesses the vegetation dynamics in Shaanxi Province from 2000 to 2019 with analyzing the vegetation response patterns to drought. The drought was calculated by the Standardized Precipitation Evapotranspiration Index (SPEI). Also, the future trends in aridification/humidification and the drought sensitivity of vegetation were predicted.

This paper covers an important subject and provides comprehensive results and discussion in terms of vegetation dynamics and its relation to drought. I believe it should be accepted with minor revision at this time. Several minor points should be modified to become much better with a higher impact. I list out some main points below and then the comments for the lines.

General comments:

1-    Introduction: It gives a clear and specific background of vegetation, drought, and literature. No need for any modification.

2-    Methods: The methodology used in this paper is clear. I just recommend explaining more about Monthly MVC. Also, the first step in the SPEI method is fitting the original data into the most suitable probability density function. So, Goodness-of-Fit tests are a significant procedure before standardization into Z scores.

3-    Drought sensitivity classification needs more discussion.

4-    Results and figures are clear and provide a comprehensive view.

Specific comments by line number:

Figure 2: Need to mention the maximum value composite in section 2.2

Line 141: Which program did you use for calculating the SPEI? For each pixel (1 km x 1 km), you have a SPEI value. Which program or code did you use?

Line 143: change it to Pi instead of PER. No need to use PRE and P. Just use them for the whole article.

Line 148: “fitted the Di sequence using a three-parameter log-logistic distribution. “ You have to check if the 3 parameters log-logistic probability distribution function is suitable for your data (D) using Goodness-of-Fit tests. 

Line 155: “SPEI = 0.5 and SPEI = -0.5 are taken as the threshold values of humidity and drought” Why why did you use these values? Please add a reference for this statement.

Line 162: “ j denotes the timescale of the SPEI from 1 to 12 months” 12 or 24? Revise equation 7.

Also, did you calculate the SPEI for 1, 2, 3, …, 12-month times scale?

Line 163 and equation 8: You have calculated the correlation coefficient for each month and for each time scale. Why did you take the maximum R-value? This point needs more discussion.

Table 2: The classification of R values with their sensitivity needs more discussion. Why did you use 0.56 as extreme sensitivity?

Line 299: As expected and proved in previous studies mentioned in the introduction.

Sincerely,

Minor editing of English language required

Round 2

Reviewer 1 Report

 Accept in present form